# Activation of the RIG-I/MAVS Signaling Pathway during Human Adenovirus Type 3 Infection Impairs the Pro-Inflammatory Response Induced by Secondary Infection with *Staphylococcus aureus*

**DOI:** 10.3390/ijms25084178

**Published:** 2024-04-10

**Authors:** Jiehan Chen, Qiaowen Wang, Biying Zhong, Huiying Zheng, Dingjun Wang, Xiao Huang, Li Liu, Tiantian Liu

**Affiliations:** School of Public Health, Guangdong Pharmaceutical University, Guangzhou 510310, China; chenjie.han@163.com (J.C.); wqw_hhh@163.com (Q.W.); zby_1232393@163.com (B.Z.); zhyzhy_2021@163.com (H.Z.); wdj1277011327@163.com (D.W.); 13142010071@163.com (X.H.)

**Keywords:** coinfection, adenovirus type 3, *Staphylococcus aureus*

## Abstract

The exacerbation of pneumonia in children with human adenovirus type 3 (HAdV-3E) is secondary to a *Staphylococcus aureus* (*S. aureus*) infection. The influence of host–pathogen interactions on disease progression remains unclear. It is important to note that *S. aureus* infections following an HAdV-3E infection are frequently observed in clinical settings, yet the underlying susceptibility mechanisms are not fully understood. This study utilized an A549 cell model to investigate secondary infection with *S. aureus* following an HAdV-3E infection. The findings suggest that HAdV-3E exacerbates the *S. aureus* infection by intensifying lung epithelial cell damage. The results highlight the role of HAdV-3E in enhancing the interferon signaling pathway through *RIG-I* (*DDX58*), resulting in the increased expression of interferon-stimulating factors like *MX1*, *RSAD2*, and *USP18*. The increase in interferon-stimulating factors inhibits the NF-κB and MAPK/P38 pro-inflammatory signaling pathways. These findings reveal new mechanisms of action for HAdV-3E and *S. aureus* in secondary infections, enhancing our comprehension of pathogenesis.

## 1. Introduction

The human adenovirus is a non-enveloped, double-stranded DNA virus that was first isolated in 1953 from the surgically removed glands of children [1]. To date, human adenoviruses are categorized into seven subgroups, A-G, with at least 111 types (Accessed on 25 December 2023 at http://hadvwg.gmu.edu/). The human adenovirus causes many diseases, and its infection of the respiratory tract is one of the main modes of prevalence. In 2016, three outbreaks of an adenovirus-related acute respiratory disease in troops occurred in the provinces of Tibet, Sichuan, and Yunnan [2]. The main adenovirus types associated with these respiratory systems are C1, C5, and C6 in subgroup C and HAdV-3, HAdV-7, and HAdV-14 in subgroup B [3]. These viruses can readily cause lung infections in the elderly and children, particularly in children under 5 years of age who are more vulnerable to a human adenoviral infection, leading to severe pneumonia [4,5]. Adenovirus pneumonia has been documented to represent around 10% of childhood pneumonias, and it can spread easily, sometimes resulting in fatal illness in healthy children during epidemics or outbreaks [6]. During epidemics or outbreaks, adenoviruses can lead to necrotizing bronchiolitis and fine bronchiolitis with extensive exfoliation of the surface epithelium. This is typically observed following a viral infection of the lungs, particularly in medium-sized intrapulmonary bronchi [7,8,9]. The bronchial and fine bronchial lamina propria are usually congested and edematous, infiltrated predominantly with mononuclear inflammatory cells. Amorphous eosinophilic material, mixed inflammatory cells, a detached epithelium, and cellular debris may occlude the affected airways, obstructing respiration [6]. Since many viruses can disrupt the airway epithelium, this may increase the likelihood of its adherence to the airways and bacterial translocation [8,10]. Additionally, viruses can lead to the dysregulation of various components of the immune system, both pro- and anti-inflammatory factors, thereby promoting pathogenesis by opportunistic pathogens such as *S. aureus*. Previous reports have also confirmed the high prevalence of coinfections during respiratory infections in children [11,12]. Among 99 children hospitalized with pneumonia in Quanzhou Women and Children’s Hospital, an adenoviral–bacterial coinfection accounted for 63.64% of the adenovirus coinfections [13]. Adenoviruses have similar susceptible populations and epidemiologic seasons as other respiratory pathogens, making adenoviruses susceptible to coinfections with other bacteria [14].

*Staphylococcus aureus* (*S. aureus*) is a common Gram-positive pathogen that is commonly secondary to viral pneumonia infections. It was first detected in 1880 in the pus of a surgical abscess [15]. Soon after the discovery of *S. aureus*, it was quickly realized that *S. aureus* is a powerful pathogen, and it remains one of the leading causes of bacterial infections globally [16]. Especially among neonates, infants, and immunocompromised children, *S. aureus* is one of the major pathogens responsible for community-acquired pneumonia in children. It is estimated that 30% of the human population has *S. aureus* strains colonizing the nostrils, skin, and gastrointestinal tract [17]. Nasal carriage is the most important route for *S. aureus* infection, with >80% of infectious isolates originating from the nasal cavity [18]. A retrospective cohort study showed 17% of intensive care unit patients had positive nasal swabs for methicillin-resistant *Staphylococcus aureus* (MRSA), and 28.6% of the patients with nasal MRSA colonization subsequently developed pneumonia [19]. Another study showed that in patients who were admitted to the intensive care unit with colonized *S. aureus*, the risk of subsequently developing pneumonia was up to 15 times higher [20]. Hence, the likelihood of the secondary infection with *S. aureus* is notably high among children and individuals with compromised immune systems. The pathogenicity of *S. aureus* is a result of the combination of its secreted extracellular substances (exotoxins, cofactors that activate host zymogens, and exonucleases) and the invasiveness of the strain itself [21]. Secreted exotoxins are highly inflammatory and can cause leukocyte death through cytolysis and clonal deletion, respectively [22]. Thrombin and glucokinase are cofactors that hijack the host coagulation system, leading to thrombosis and disseminated intravascular coagulation. Exonucleases cleave and inactivate various immune defense and surveillance molecules, such as complement factors, antimicrobial peptides, and surface receptors important for leukocyte chemotaxis, which also contribute to pathogenesis. In addition, some secreted toxins and extracellular enzymes lead to the disruption of endothelial and epithelial barriers through cell lysis and the cleavage of connexins [23]. Following a viral infection accompanied by the necrosis of ciliated and ringed cells, the *S. aureus* adhering to the lower respiratory tract can shift from colonization to pneumonia. A secondary infection caused by bacteria may also further promote a viral infection by altering the ease with which viruses can be transmitted and infected within the respiratory system, from which they act as mutually reinforcing infections [10].

Children with adenovirus infection in their respiratory tract often experience co-infection with *S. aureus*, leading to severe pneumonia. A surveillance report on community-acquired pneumonia coinfections in China from 2009 to 2020 highlighted the significant association between community-acquired pneumonia in children and adolescents, and co-infection with adenovirus and *S. aureus* [24]. While there has been progress in understanding the molecular mechanisms of adenovirus and *S. aureus* infections individually, the molecular mechanism of pneumonia resulting from secondary *S. aureus* infection following adenovirus infection remains poorly understood. To address this gap, we established a co-infection cell model of HAdV-3E and *S. aureus,* and utilized high-throughput sequencing methods to elucidate the interaction between adenovirus and *S. aureus* on lung epithelial cells. Our study identified the importance of RIG-I targets in bacterial infections, laying the foundation for potential drug development targeting RIG-I in adenoviral-bacterial coinfections.

## 2. Results

### 2.1. Effects of HAdV-3E with Secondary S. aureus Infection on Epithelial Cell Damage at Different Time Points

The model was developed by subjecting cells to continuous infection with HADV-3E at varying MOI for 48 h, followed by the secondary infection with *S. aureus* at different MOIs for durations of 6 h, 9 h, and 12 h. Results indicated that in the 9-h and 12-h models of secondary infection with *S. aureus*, cell survival rates in the secondary infection group were consistently below 60%, reflecting significant cell damage and low viability, rendering them unsuitable for further experimentation (Appendix A).

In the model where cells were infected with HAdV-3E for 48 h followed by secondary infection with *S*. *aureus* for 6 h, a MOI of 10^−3^ for HAdV-3E and a MOI of 10^−1^ for *S*. *aureus* resulted in a cell survival rate of 61.96%. This rate was lower than the 67.62% cell survival rate observed in the HAdV-3E alone infection group and the 78.00% cell survival rate in the *S. aureus* alone infection group. This model not only mirrors the clinical scenario where coinfection exacerbates cell damage but also maintains the necessary cell viability for subsequent experiments (Figure 1).

Epithelial cells were examined using fluorescence microscopy in a model of HAdV-3E infection in the A549 cell line over a period of 24 h, 36 h, and 48 h, followed by a secondary infection with *S. aureus* for 6 h in a coinfection scenario. The results show that after 24 h of coinfection, some cells displayed significant lesions and rounding in the bright field images, although no aggregation was observed. Specifically, the 24 h coinfection group (A3SA_24) exhibited partially rounded lesions under a bright field, with 20% of the epithelial cells expressing green fluorescence, which was indicative of an infection. In the 36 h coinfection group (A3SA_36), the cell lesions were more pronounced, leading to cell aggregation and syncytia formation. Approximately 50–60% of the cells were fluorescing in the fluorescent field. In the 48 h coinfection group (A3SA_48), cell fusion was further pronounced and accompanied by cellular senescence, with over 70% of the cells fluorescing. The findings suggest that as the duration of the adenoviral infection increases, the damage to lung epithelial cells escalates, ultimately leading to necrosis (Figure 2).

### 2.2. Effects of Different Durations of HAdV-3E Infection on the Expression of Coinfected Genes

HAdV-3E was used to infect the A549 cell line for 24 h, 36 h, and 48 h, followed by a 6 h *S. aureus* infection, resulting in the formation of A3SA_24, A3SA_36, and A3SA_48. This created three infection groups and nine datasets. A hierarchical cluster analysis was conducted on the three groups of data to evaluate the data quality. The results, depicted in Figure 3A, show that A3SA_24, A3SA_36, and A3SA_48 were clustered together in one category, indicating good data quality across all three groups.

The coinfection data for 24 h, 36 h, and 48 h HAdV-3E infections were analyzed for the differential genes. Screening conditions of |log_2_FC| > 0.2 and q-value < 0.05 were applied. The results revealed the highest number of Significant differentially expressed genes (DEGs) in A3SA_48 compared to A3SA_24, with 1825 DEGs. Of these, 906 were down-regulated and 919 were up-regulated genes. In A3SA_48 vs. A3SA_36, there were 865 DEGs, with 496 down-regulated and 369 up-regulated genes. Lastly, in the A3SA_36 vs. A3SA_24, there were 97 DEGs, with 51 down-regulated and 46 up-regulated genes (Figure 3B–D).

The results of the volcano plots were analyzed to compare the differential gene expressions between A3SA_48 and A3SA_24, A3SA_48, and A3SA_36, as well as A3SA_36 and A3SA_24. DEGs were observed, indicating a potential correlation with the varying timeline of HAdV-3E infection in A549 cells followed by a secondary infection with *S. aureus*. This suggests an impact on the host cell’s resistance to both the virus and the bacterium, highlighting significant differences in the host cell responses at different stages of viral infection.

With a prolonged infection duration, there was a noticeable increase in the up-regulation of the expression of RIG-I-like receptor gene (*RIG-I*/*DDX58*) in the three-group comparison (log_2_FC > 0.2, q-value < 0.05). This indicates that the up-regulation of *RIG-I* may enhance the interferon signaling pathway during extended viral infections. Additionally, key genes involved in the IFN signaling pathway such as *IFNGR1*, *JAK*, *STAT*, and *IRF* showed a tendency towards up-regulation.

The results of differential gene analysis of all groups showed that IFN signaling pathway was enhanced and the expression of interferon-stimulating genes (ISGs) such as *MX1*, *RSAD2*, *USP18* and *ISG15* was up-regulated. The up-regulation of these genes in a temporal sequence may potentially contribute to secondary bacterial infections. Furthermore, *IRAK1*, a significant regulator of IL-1R- and TLR-mediated signaling, exhibited a trend of down-regulation.

The *MX1* gene exhibited a notable up-regulation in all three groups compared to the controls (log_2_FC > 0.2, q-value < 0.05). Within host cells, the Mx protein plays a crucial role in enhancing the antiviral effects of IFN-I, which in turn stimulates Mx protein production. Moreover, MxA exerts inhibitory effects on the NF-κB pathway. Through temporal transcriptome sequencing, we observed a decreasing trend in *RelA* (*P65*), a key component of the classical NF-κB pathway, over time, although it was not statistically significant (log_2_FC < −0.2, q-value > 0.05). This down-regulation subsequently hinders the generation of antimicrobial inflammatory factors and facilitates the *S. aureus* infection.

*RSAD2*, an ISGs activated by RIG-I signaling through the IFN-I pathway, plays a crucial role in the innate antiviral response. In a comparative analysis involving three groups, *RSAD2* demonstrated a significant increase in its expression levels at A3SA_48 compared to both A3SA_36 and A3SA_24 h (log_2_FC > 0.2, q-value < 0.05). This protein class, induced by interferons, exhibits antiviral properties by inhibiting viral replication. Furthermore, the literature indicates that RSAD2 can enhance a *Mycobacterium tuberculosis* infection by negatively regulating IRAK1-TRAF6-TAK1 signaling [25]. Additionally, our temporal transcriptome analysis revealed a notable decrease in the expression levels of *IRAK1* and *MAPK11* (*P38-2*) over time, with significant differences observed particularly between A3SA_48 and A3SA_24 h (log_2_FC > 0.2, q-value < 0.05). The down-regulation of *IRAK1* may lead to the negative regulation of the NF-κB classical pathway and the MAPK P38 pathway, which, in conjunction with *MX1*, suppresses the production of antimicrobial inflammatory factors, thereby promoting the *S. aureus* infection while inhibiting viral replication.

A further analysis of the differentially expressed genes revealed an increase in the negative regulator *USP18*, a target of RIG-I in the IFN-I and IFN-III pathways, showing significance in the A3SA_48 vs. A3SA_24 h comparison. USP18 can hinder IFN-I signaling by disrupting the JAK1-STAT2 interaction at the IFNAR2 receptor, thus promoting viral replication. However, USP18 also plays a role in enhancing the antiviral effects of the interferon signaling pathway by modulating the cGAS-STING and TRIM31-MAVS pathways. In bacterial infections, USP18 inhibits NEMO ubiquitination, negatively regulating the TAK1-TAB complex to suppress NF-κB activation. Additionally, USP18 dampens antibacterial TNF-α signaling and ROS production, increasing the susceptibility to bacterial infections. The up-regulation of *ISG15*, closely linked to USP18 stability, was also observed in the temporal transcriptome. ISG15, an IFN-I- and IFN-III-induced ubiquitin-like protein, has a controversial role in the IFN-I pathway, but its consistent increase is crucial for maintaining USP18 stability and function.

### 2.3. Effects of Different Durations of HAdV-3E Infection on Gene Expression Trends in Coinfected Cells

The enrichment of all transcribed genes using Muffz software (Version 3.18 created by Matthias Futschik) in R language revealed six distinct clusters with varying trends, as depicted in Figure 4B showed the enrichment of NF-κB and MAPK pro-inflammatory pathway related genes, including *IRAK1*, *RELA*, *ICAM1*, and *MAPK11*. These genes exhibited a tendency towards down-regulation following an increased duration of the viral infection, particularly between 24 to 36 h post-infection, with a more pronounced decrease in the expression levels.

In Figure 4C, the enrichment of antiviral-related genes such as *DDX58*, *RSAD2*, *USP18*, *STAT1*, and *JAK1* was observed. This suggests that the antiviral response is significantly enhanced as the duration of the viral infection increases.

Seven differentially expressed genes (*DDX58*, *STAT1*, *MX1*, *IRAK1*, *MAPK11*, *RELA*, and *ICAM1*) were randomly selected and quantified using qRT-PCR to validate the differential expression observed in A549 cells using RNA-Seq across varying infection time points. The qRT-PCR results indicated an up-regulation trend for *DDX58*, *STAT1*, and *MX1*, while *IRAK1*, *MAPK11*, *RELA*, and *ICAM1* exhibited a down-regulation trend as the duration of the viral infection increased (Figure 5B). These findings were consistent with the patterns observed in high-throughput RNA sequencing (Figure 5A).

### 2.4. Enrichment Analysis Suggests That an Increased Duration of HAdV-3E Infection Promotes the Antiviral Pathway and Inhibits the Inflammatory Pathway

A Gene Ontology functional enrichment analysis (GO enrichment) was conducted on the differentially expressed genes in the A3SA_48 vs. A3SA_24, A3SA_48 vs. A3SA_36, and A3SA_36 vs. A3SA_24 comparisons. Figure 6, Figure 7 and Figure 8 illustrate the top ten GO pathways that were enriched and significantly differentially expressed (q-value < 0.05) in the three datasets.

The comparison between the A3SA_48 and A3SA_24 groups revealed a significant enrichment in cellular components such as ‘cytoplasm (GO:0005737)’ and ‘organelle (GO:0043226)’, as shown in Figure 6. Similarly, the comparison between A3SA_48 and A3SA_36 showed an enrichment in ‘cytoplasm (GO:0005737)’ and ‘ribosome (GO:0005840)’, as depicted in Figure 8. This enrichment may be attributed to the increased expression of ISGs like *MX1* and *RSAD2* during the early stages of the viral infection, leading to a higher demand for enzymes and energy production in organelles such as mitochondria. Additionally, the ubiquitination of IFN-I by USP18 primarily occurs in mitochondria, further supporting the enrichment of genes related to this organelle. The positive feedback loop of interferon-stimulated genes results in an increased synthesis of antiviral proteins, subsequently inhibiting viral protein translation by ribosomes and impeding viral replication.

Figure 9, Figure 10 and Figure 11 display the top 20 KEGG pathways that exhibit significant differences in expression (q-value < 0.05) across the three datasets. Among these pathways, the ‘Influenza A pathway (ko05164)’ was notably enriched and showed a significant enhancement in A3SA_48 vs. A3SA_24 and A3SA_48 vs. A3SA_36 (Figure 9 and Figure 10). The interferon pathway, regulated by the IFNGR receptor, was also enhanced, leading to the increased production of ISGs like *MX1*, *RSAD2*, and *USP18*, which play a role in inhibiting viral replication.

In the comparison between A3SA_48 and A3SA_24, we observed the inhibition of the signaling pathway related to ‘American trypanosomiasis (ko05142)’ (Figure 9). Additionally, in the A3SA_36 vs. A3SA_24 comparison, we noted the down-regulation of *MyD88*, a regulator of NF-κB, in ‘malaria (ko05144)’ and ‘shigellosis (ko05131)’, leading to the inhibition of the antibacterial NF-κB pathway (Figure 11).

The GSEA enrichment analysis revealed that the *S. aureus* infection pathway (ko05150) exhibited significant activation in both the infected A3SA_48 and A3SA_24 groups, as well as in the A3SA_36 and A3SA_24 groups (nominal *p*-value < 0.05, FDR < 0.25, normalized ES > 1) (see Appendix A). This suggests that during the middle and late stages of an adenoviral infection, cells experience an immune inflammation activation following *S. aureus* infection, albeit to a certain extent of suppression. This implies a worsening of the *S. aureus* infection pathway (KO5150) post-bacterial infection.

In our analysis of the genomic enrichments during the late stage of infection in A3SA_48 and A3SA_36, we observed an enrichment of the NOD-like receptor signaling pathway (KO4621) and the influenza A-enriched pathway (KO5164), both of which are linked to a viral infection. (Appendix A). This suggests that cells are more significantly impacted by the virus during the later stages of infection, with an enhanced IFN-I effect induced by RIG-I.

### 2.5. The Inflammatory Pathway Induced by RIG-I and the Staphylococcus aureus Infection Pathway Coregulate the Production of Inflammatory Factors

It was observed that *RIG-I* plays a central role in the antiviral and inflammatory pathways, regulating various inflammatory pathways such as NF-κB, MAPK, and JAK/STAT (Figure 12). *S. aureus* has the ability to hinder LTA1-ICAM-1 action by binding its own Eap to the ECM or ICAM-1, thus impeding neutrophil and T-cell recruitment, inhibiting T-cell proliferation, and ultimately, suppressing immune responses. In addition, activation of the RIG-I receptor can stimulate the RIG-I pathway, leading to increased IFN production. This in turn activates the JAK/STAT pathway, influencing the production of ISGs, which ultimately impacts the antiviral response and the production of inflammatory factors.

### 2.6. The RIG-I Signaling Pathway Inhibits the Activation of Inflammatory Pathways

Following the inhibition of the RIG-I signaling pathway in A549 cells with the RIG012 inhibitor, the cells’ transcriptome was analyzed using qRT-PCR. The results showed a significant decrease in *MX1* gene expression compared to the coinfected group, and a notable increase in the RIG-I suppression group at 24 h, 36 h, and 48 h (Figure 13A–C). Additionally, the *RELA* gene exhibited a significant increase in the suppression group compared to the coinfected group (Figure 13D–F). These findings suggest that inhibiting the RIG-I pathway can suppress the interferon-stimulating factor of the IFN-I signaling pathway at the mRNA level, while simultaneously enhancing the NK-κB pro-inflammatory pathway significantly.

## 3. Discussion

Epithelial cells play a crucial role in the host’s immune response by producing cytokines and chemokines to combat pathogens. This study aimed to investigate the changes in gene expression in lung epithelial cells during a secondary infection with *S. aureus* at various stages of viral infection. By creating a model of secondary infection with *Staphylococcus aureus* following different durations of viral infection and utilizing RNA-Seq technology, we explored uncharted territory, as there has been no prior research on secondary infection with *S. aureus* after adenovirus infection of epithelial cells. This study breaks new ground by utilizing in vitro transcriptomics to elucidate the mechanisms of epithelial cells in response to a bacterial infection during different stages of an adenoviral infection.

RIG-I is an innate immune receptor that plays a crucial role in the interferon signaling pathway by detecting viral nucleic acids in the cytoplasm and triggering a signaling cascade that boosts the production of IFN-I, ultimately promoting ISGs production [26]. Studies have shown that ISGs production can facilitate bacterial infections. For example, the up-regulation of RSAD2 in *Mycobacterium tuberculosis* hinders host defense activation and antigen presentation in dendritic cells during infection with this pathogen [27]. Conversely, the down-regulation of MxA leads to an increased production of inflammatory cytokines such as IL-1β, IL-6, and TNF-α, along with NF-κB p65 activation, resulting in a decreased intracellular *Mycobacterium tuberculosis* infection [25]. USP18 has dual effects on IFN-I signaling regulation, potentially promoting bacterial replication by suppressing TNF-α signaling [28]. Furthermore, as the adenoviral infection progresses, there is a noticeable enhancement in the IFN-I signaling pathway and an increase in interferon-stimulating factor levels. Simultaneously, there is a partial inhibition of inflammatory pathways like NF-κB and MAPK/P38, which can hinder bacterial proliferation.

*RELA* is a crucial transcription factor in the NF-κB signaling pathway. Our study revealed that as viral infection progresses, despite all three time series remaining active, the transcription of *RELA* gradually decreases. This suggests that the NF-κB classical pathway in epithelial cells during a secondary *S. aureus* infection tends to diminish as the viral infection duration increases, despite being in an activated state. Similar results were observed in a mouse model of influenza following *Staphylococcus aureus* infection [29]. MAPK11, a member of the p38 MAPK family, plays a significant role in the cellular response to extracellular stimuli like inflammatory responses [30]. The decrease in *RELA* and *MAPK11* indicates a tendency towards dampening some inflammatory pathways in the epithelium following a secondary *S. aureus* infection as the adenoviral infection worsens. Furthermore, there is a noticeable reduction in *IRAK1* during this process. IRAK1 is a critical gene in Toll-like receptor (TLR) and IL-1R signaling pathways, triggering the activation of NF-κB and IFN signaling pathways. The diminishing *IRAK1* levels over time during the viral infection suggest a negative feedback mechanism in response to antiviral and antibacterial infections, potentially enhancing the *S. aureus* infection by modulating downstream inflammatory pathways.

The enrichment pathway analysis revealed the enhanced control of adenoviral infection within the cell, leading to an increase in various biological processes related to viral proliferation. These processes include the viral defense response, viral life cycle, viral gene expression, and viral transcription. Additionally, the KEGG-enriched pathway indicated a significant intensification of the influenza A infection pathway, suggesting a worsening viral infection. Furthermore, the presence of American trypanosomiasis, malaria, and shigellosis in the KEGG analysis implied ongoing antimicrobial resistance that tended to decrease with a prolonged viral infection. This indicates that as the adenoviral infection progresses, *S. aureus* may experience stress and release complement factors like CFH and CFB into epithelial cells to prevent the over-activation of C3 and evade the immune response [31,32,33]. Moreover, *S. aureus* autoprotein Eap can block adhesion factors such as ICAM-1, inhibiting leukocyte recruitment [33].

Our study suggests that RIG-I is an important target for regulating the effects of coinfection, especially for promoting bacterial proliferation. It does this by inducing the formation of interferons, and thus, the production of interferon-stimulating factors. Interferon-stimulating factors such as *RSAD2*, *MX1*, *USP18*, etc., reduce the production of inflammatory factors by indirectly inhibiting inflammatory signaling pathways such as NF-κB, MAPK/P38, etc., which reduces the recruitment of immune cells and allows Aureobasidium to evade immune responses. We also demonstrate that intervening with anti-RIG-I during the early stages of an adenoviral infection in vivo thereby reduces the severity of secondary infections with *S. aureus* and provides a therapeutic strategy for viral–bacterial coinfection.

## 4. Materials and Methods

### 4.1. Viruses, Cells, and Bacteria

The recombinant adenovirus type 3 (HAdV-3E) containing enhanced green fluorescent protein in this study was kindly gifted by Prof. Rong Zhou (State Key Laboratory of Respiratory Diseases, Guangzhou, China). Adenoviruses were cultured in adenocarcinomic human alveolar basal epithelial cells (A549) that were obtained from the American Type Culture Collection (Manassas, VA, USA) and subsequently maintained in our lab. The biocamera microscope used to observe the cells and fluorescence is a Leica DM IL LED (Leica, Wetzlar, Lahn-Dill-Kreis, Germany). RIG-I inhibitor is RIG012 (Axon, Groningen Netherland). A549 cells were cultured in Dulbecco’s Modified Eagle’s Medium (DMEM; Gibco, Carlsbad, CA, USA)) containing 2% Penicillin-Streptomycin (Gibco, Carlsbad, CA, USA)) and 10% fetal bovine serum (Invitrogen, Carlsbad, CA, USA). The standard strain of *Staphylococcus aureus* ATCC 25923 (*S. aureus*) was obtained from the Guangdong Food Safety Strain Preservation Center and cultured in LB liquid medium or nutrient agar (Huankai Biology, Guangzhou, China).

### 4.2. Modeling Coinfection

A549 cells were cultured in 6-well plates using 2% FBS antibiotic-free DMEM according to cell growth conditions. When the cell growth reached about 70–80% (5 × 10^5^ cells/well), the A549 cells were infected with HAdV-3E with an MOI = 0.001 for 24 h, 36 h, and 48 h, followed by a secondary infection of the A549 cells with *S. aureus* with an MOI = 0.1 for 6 h (Figure 14).

### 4.3. RNA Extraction, Library Construction, and Sequencing

After washing the treated sequencing cell samples 3 times with PBS, the total RNA was isolated from the coinfected A549 cells using the TRIzol™ Plus RNA Purification Kit (Thermo Fisher Scientific, Carlsbad, CA, USA) according to the manufacturer’s instructions. The purity and concentration of the RNA were determined using a Merinton SMA4000 (Merinton, Beijing, China) and a Qubit2.0 Fluorometer (Thermo Fisher Scientific, Carlsbad, CA, USA). After the RNA purity and concentration were established, DNA (cDNA) libraries were constructed using the Hieff NGS™ MaxUp Dual-mode mRNA Library Prep Kit for Illumina^®^ (Yeasen, Shanghai, China) according to the manufacturer’s instructions. The library was amplified and constructed through the steps of purifying and fragmenting mRNA, synthesizing and purifying double-stranded cDNA, end repair/dA tail addition, the ligation of adapters, purification of ligation products, and amplification. Gel electrophoresis was used to detect the library. The detected library was between 300–500 bp and was considered a qualified library. Finally, the qualified libraries were subjected to paired-end sequencing using an Illumina HiSeq4000 (Illumina, San Diego, CA, USA).

### 4.4. Sequencing Data Preprocessing

Before alignment, FastQC was used to check the quality of the raw reads generated using the Illumina Hiseq 4000 platform, which were filtered by removing the dirty raw reads. Reads containing an adapter, unknown base > 10%, and low-quality reads (the base number of a threshold mass ≤ 10 accounts for more than 50% of the total reading) were removed to obtain clean reads of mRNA. The sequences that were sequenced after quality control were compared with the human reference genome (GRCh38) using HISAT2, and the comparison results were counted using RSeQC.

### 4.5. Analysis of Genetic Differences

The expression levels of the gene transcripts were calculated using StringTie, and then, DEGseq was used for the differential gene (DEGs) analysis using a q-value < 0.05 and multiplicity of differences |FoldChange| > 1.5. The differential genes were visualized by drawing volcano plots using ggplot2 in the R package.

### 4.6. qRT-PCR to Detect Genetic Trends

The accuracy of the RNA-Seq results was verified using qRT-PCR on randomly screened differential genes. The total RNA was isolated from the cells using Trizol reagent, and the cDNA was removed and reverse transcribed to form cDNA using a reverse transcription kit (Toyobo, Osaka, Japan). Primers and SYBR Premix Ex Taq II (Takara, Kyoto, Japan) were added for 30 s at 95 °C, followed by 40 cycles of PCR at 95 °C for 5 s and at 60 °C for 34 s. This procedure was carried out using an ABI7500 instrument (Applied Biosystems, Foster City, CA, USA), and the results were expressed using the 2^−∆∆Ct^ method for representation.

### 4.7. Gene Enrichment Analysis

DEGs were included in the GO (Gene Ontology) database (Accessed on 25 August 2023 at https://www.geneontology.org/) and KEGG (Kyoto Encyclopedia of Genes and Genomes) database (Accessed on 25 August 2023 at https://www.genome.jp/kegg/) for enrichment analysis. Three types of molecular function, cellular component, and biological process pathway enrichments and one type of KEGG pathway enrichment were obtained with a q-value < 0.05.

All genes were included in the GESA database (Accessed on 25 August 2023 at https://www.gsea-msigdb.org/gsea/index.jsp) for enrichment, and a nominal *p*-value < 0.05, FDR < 0.25, and normalized ES > 1 were used as the conditions for enrichment to the obtain antiviral, inflammatory, and *S. aureus*-related pathways.

### 4.8. STRING Protein Interaction Network Analysis

The key genes of RIG-I-induced antiviral and inflammatory pathways and genes related to *Staphylococcus aureus* infections were included in the STRING database to construct an interoperability network and were mapped using Cytoscape. The darker the color of the genes meant a stronger association role in the network.

## Figures and Tables

**Figure 1 ijms-25-04178-f001:**
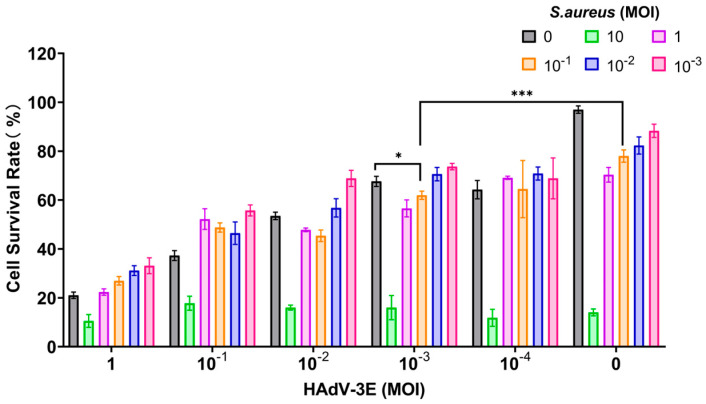
Infection with HAdV-3E 48 h coinfected with *S. aureus* infection for 6 h (* indicates *p* < 0.05, and *** indicates *p* < 0.001).

**Figure 2 ijms-25-04178-f002:**
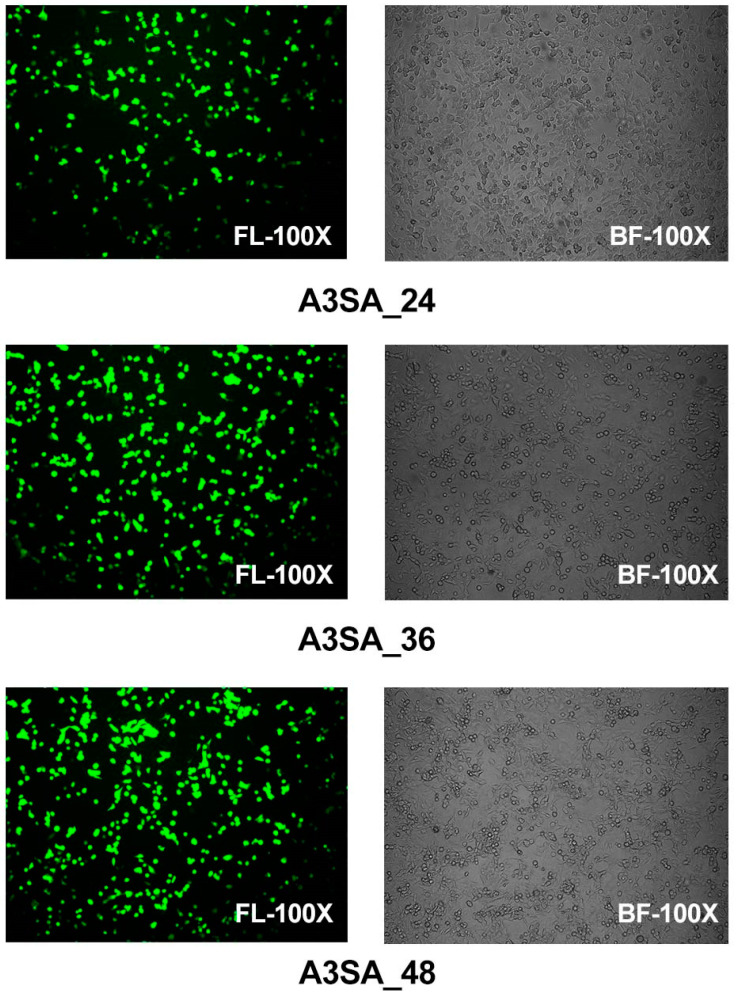
Visual fields of HAdV-3E in a bright field (BF) and fluorescent field (FL) at 100 times for 24 h, 36 h, and 48 h after infection of the A549 cell line, with a 6 h secondary infection with Aureus spp.

**Figure 3 ijms-25-04178-f003:**
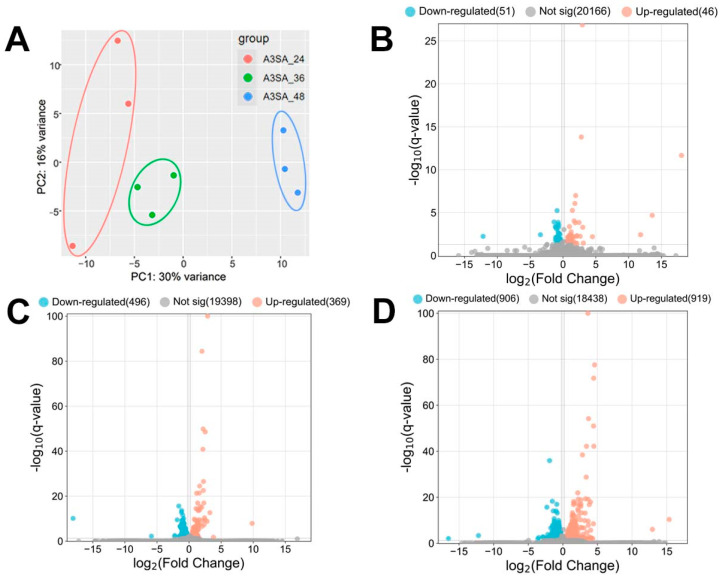
(**A**) PCA maps and (**B**–**D**) volcano maps of host cell genes between the 3 groups of samples (**B**: A3SA_36 vs. A3SA_24; **C**: A3SA_48 vs. A3SA_36; **D**: A3SA_48 vs. A3SA_24).

**Figure 4 ijms-25-04178-f004:**
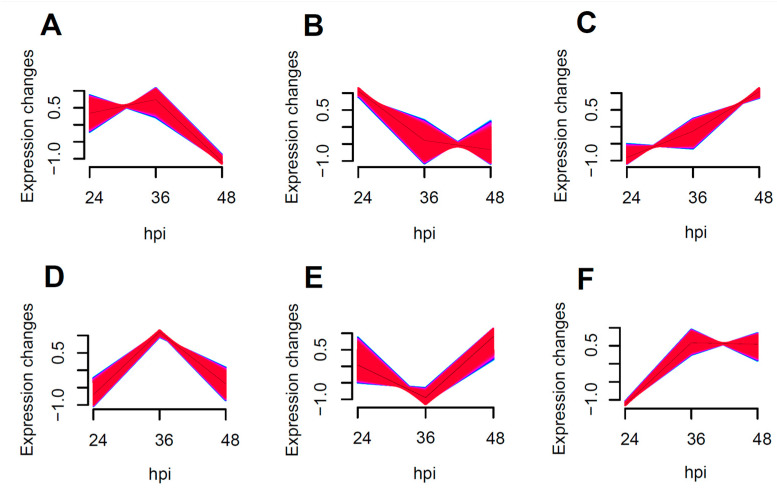
Muffz time series cluster plots for 24–48 h coinfections (Genes enriched into (**A**–**F**) 6 clusters of change graphs according to a trend of similar variation).

**Figure 5 ijms-25-04178-f005:**
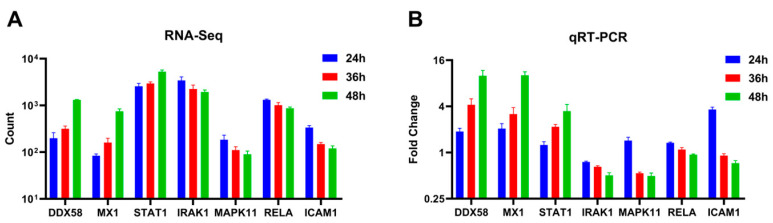
Expression levels of selected genes were quantified using (**A**) RNA-Seq and (**B**) qRT-PCR (adjusted for ACTB as an internal reference gene, expressed as a fold change, *n* = 3).

**Figure 6 ijms-25-04178-f006:**
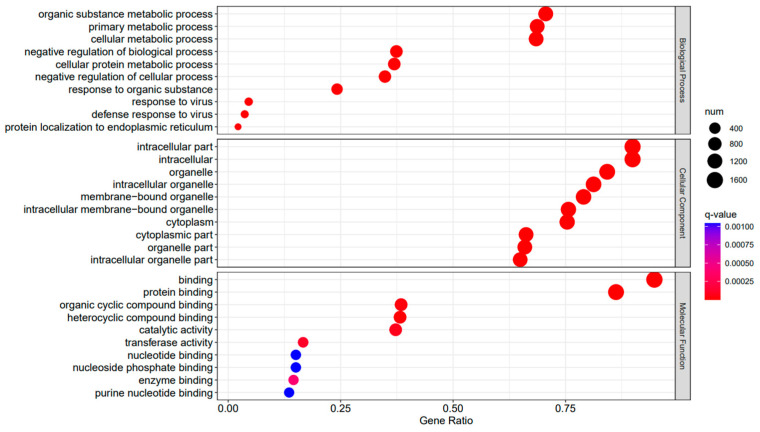
A3SA_48 vs. A3SA_24 GO enrichment analysis bubble plot. (The size of dots represents the number of enriched ontology genes. The redder the color is, the stronger the significance, and the further to the right, the stronger the gene correlation).

**Figure 7 ijms-25-04178-f007:**
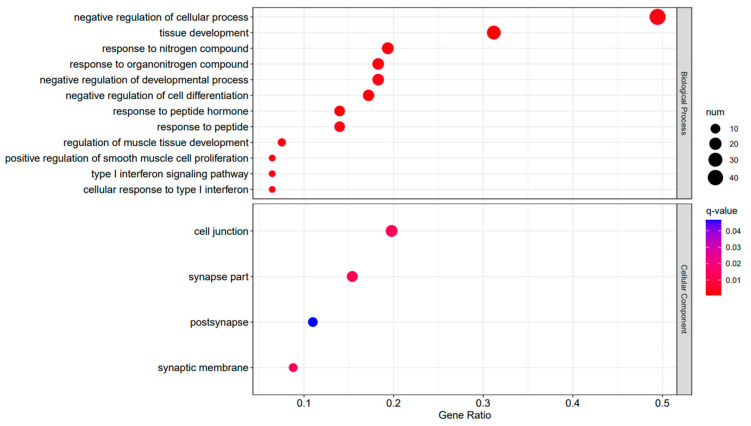
A3SA_36 vs. A3SA_24 GO enrichment analysis bubble plot. (The size of dots represents the number of enriched ontology genes. The redder the color is, the stronger the significance, and the further to the right, the stronger the gene correlation).

**Figure 8 ijms-25-04178-f008:**
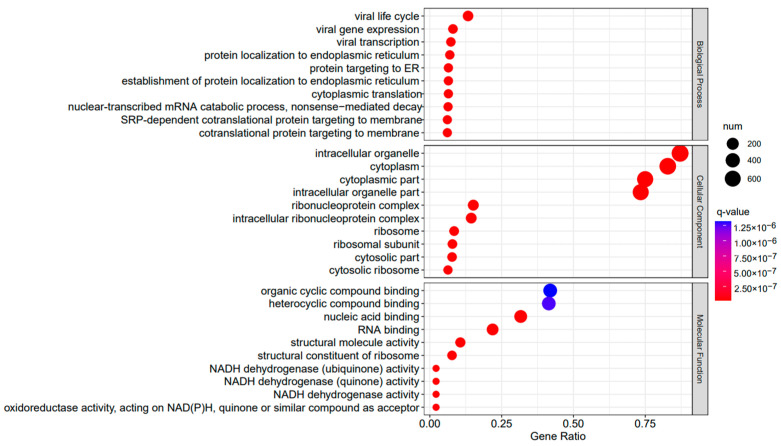
A3SA_48 vs. A3SA_36 GO enrichment analysis bubble plot. (The size of dots represents the number of enriched ontology genes. The redder the color is, the stronger the significance, and the further to the right, the stronger the gene correlation).

**Figure 9 ijms-25-04178-f009:**
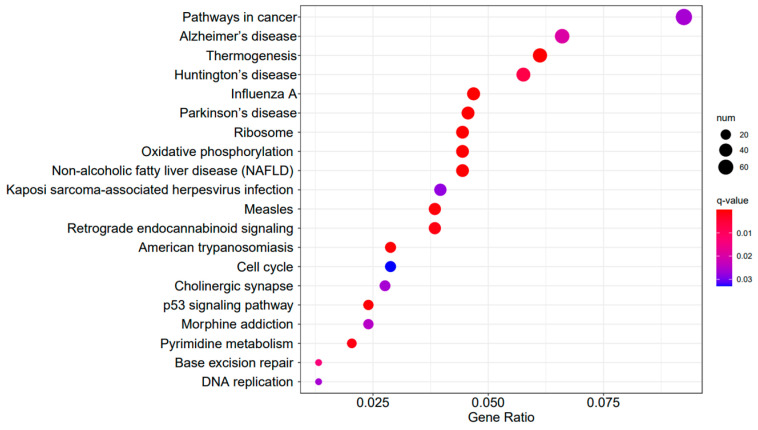
A3SA_48 vs. A3SA_24 KEGG enrichment analysis bubble plot. (The size of dots represents the number of genes enriched in the pathway. The redder the color is, the stronger the significance, and the further to the right, the stronger the gene correlation).

**Figure 10 ijms-25-04178-f010:**
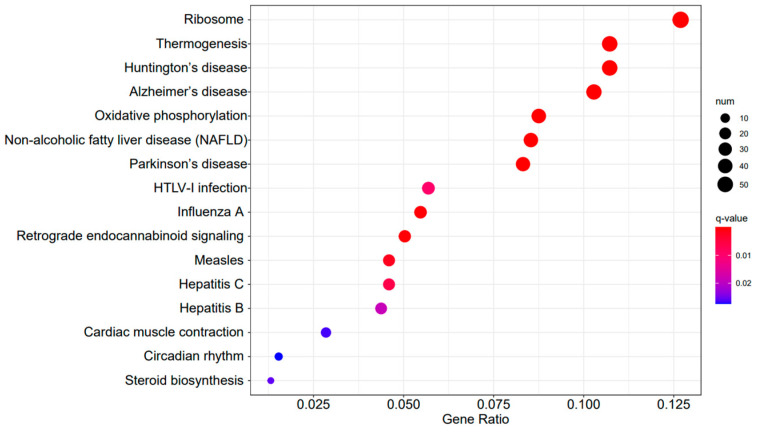
A3SA_48 vs. A3SA_36 KEGG enrichment analysis bubble plot. (The size of dots represents the number of genes enriched in the pathway. The redder the color is, the stronger the significance, and the further to the right, the stronger the gene correlation).

**Figure 11 ijms-25-04178-f011:**
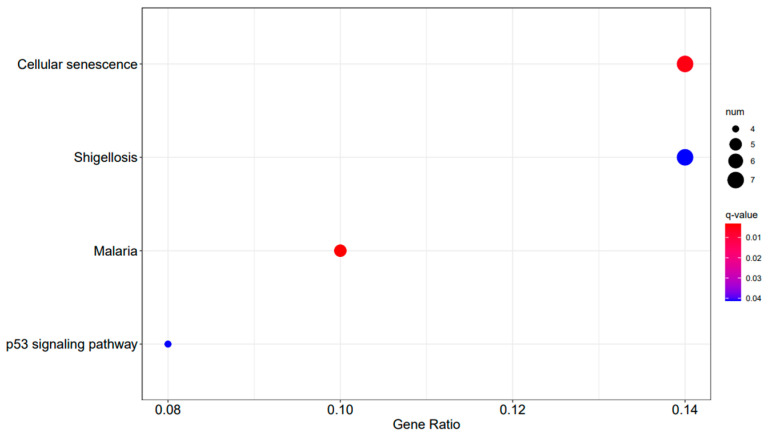
A3SA_36 vs. A3SA_24 KEGG enrichment analysis bubble plot. (The size of dots represents the number of genes enriched in the pathway. The redder the color is, the stronger the significance, and the further to the right, the stronger the gene correlation).

**Figure 12 ijms-25-04178-f012:**
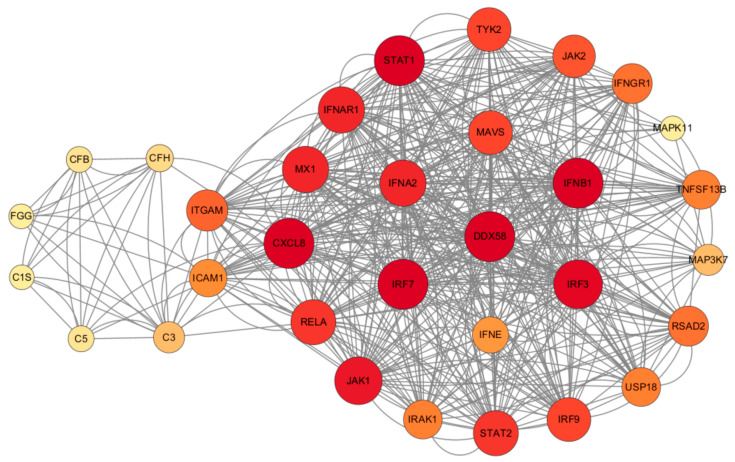
Regulatory network map of STRING interactions between *Staphylococcus aureus* infection and HAdV-3E infection-associated proteins.

**Figure 13 ijms-25-04178-f013:**
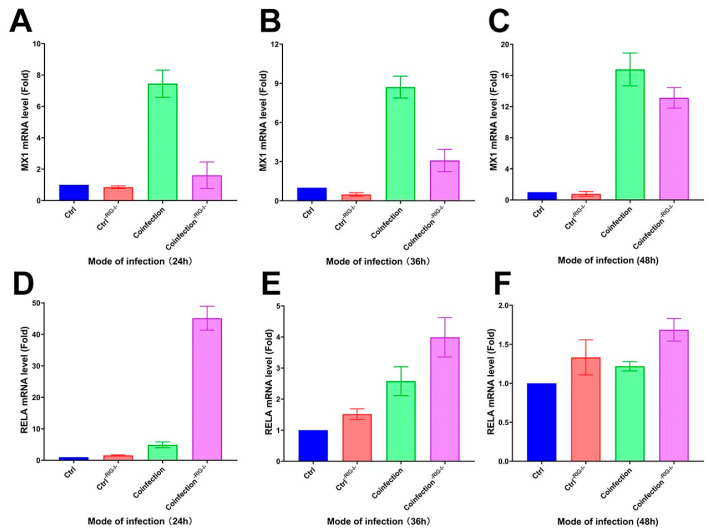
(**A**–**C**) *MX1* and (**D**–**F**) *RELA* transcriptome expression levels in RIG-I inhibitor and coinfection groups. (-RIG-I- indicates the addition of RIG-I inhibitor; (**A**,**D**) are the groups infected with HAdV-3E for 24 h and secondary infected with *S. aureus* for 6 h; (**B**,**E**) are the groups infected with HAdV-3E for 36 h and secondary infected with *S. aureus* for 6 h; (**C**,**F**) are groups infected with HAdV-3E for 48 h and secondary infected with *S. aureus* for 6 h).

**Figure 14 ijms-25-04178-f014:**
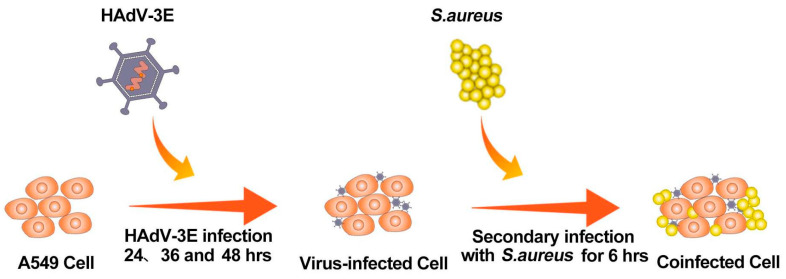
Flowchart for building a coinfection model.

## Data Availability

Data are contained within the article and Appendix A.

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
