# Peer review of "Activation of the RIG-I/MAVS Signaling Pathway during Human Adenovirus Type 3 Infection Impairs the Pro-Inflammatory Response Induced by Secondary Infection with Staphylococcus aureus"

_ijms, 2024, doi:10.3390/ijms25084178_

Round 1
Reviewer 1 Report
Comments and Suggestions for Authors
Even after multiple readings, understanding the primary aim of the manuscript remains challenging for this reviewer. The work suffers from both language deficiencies and notably poor writing quality. The presentation of data lacks clarity, and the description of results is misleading. For example, the authors demonstrated that the cells in A3SA_36 underwent necrosis. However, no evidence of necrosis was provided by the authors. It's important to note that various types of cell death might occur in response to infection. Furthermore, beyond these concerns, the experimental design lacks scientific rigor and cannot provide any conclusive insights into the interaction between HAdV-3E and S. aureus. I am sorry I have no further comments.
Comments on the Quality of English LanguageThe English language of the manuscript is of poor quality.
Author Response
Dear Reviewer:
Thanks to the reviewers for their comments and suggestions. I will comment or make changes based on your suggestions. You can see the attached “ijms-2818772-2.25” with the revisions highlighted in blue.
We have polished the language of the article to make it more fluent. We've made changes to the misleading description. At the same time, we used the existing experimental basis to infer the effects of adenovirus and Staphylococcus aureus on cells from the cellular level.
Eg. Line168-181:
Epithelial cells were examined using fluorescence microscopy in a model of HAdV-3E infection of the A549 cell line over a period of 24h, 36h, and 48h, followed by secondary infection with S. aureus for 6h in a coinfection scenario. Results showed that after 24h of coinfection, some cells displayed significant lesions and rounding in bright field, although no aggregation was observed. Specifically, the 24h coinfection group (A3SA_24) exhibited partially rounded lesions under bright field, with 20% of epithe-lial cells expressing green fluorescence indicative of infection. In the 36h coinfection group (A3SA_36), cell lesions were more pronounced, leading to cell aggregation and syncytia formation. Approximately 50-60% of cells were fluorescing in the fluorescent field. By the 48h coinfection group (A3SA_48), cell fusion was further pronounced, accompanied by cellular senescence, with over 70% of cells fluorescing. The findings suggest that as the duration of adenovirus infection increases, damage to lung epithe-lial cells escalates, ultimately leading to necrosis.
We use the degree of damage to describe the state of the cells to make the results more objective.

Reviewer 2 Report
Comments and Suggestions for Authors
The authors attempted to screen the target RIG-I for 90 the interaction of adenovirus and S. aureus coinfection by comparing high-throughput sequencing with a model of infection with HAdV-3E of different timings secondary to S. aureus. This study attempts to lay the foundation for the subsequent development of adenoviral-bacterial coinfection drugs with RIG-I targets. This is very commendable work. However, the authors need to address the following issues:
1. The overall structure of the manuscript is confusing. The methodology should come before the results to promote coherence.
2. The methodology is not very elaborately explained and the authors are not clear on the various sources of the methodology used. The authors should acknowledge the sources or if they have invented them also say so.
3. The results are not well sorted and appear very crowded, the authors should present the key results well elaborated and the rest should be appendices.
4. The discussion though well crafted is sometimes confusing, for example line 385-386 ''This suggests that as adenoviral infection worsens, S. aureus can also be stressed can also cause the release of complement factors such as CFH and CFB in the epithelial cells, which inhibit the over-activation of C3 to allow S. aureus to evade the immune response'' it is not clear whether the implication is that it this process inhibits or promotes S. aureus' evasion of the immune system. Please as you are doing English translation or editing, take care to retain the meaning.
Comments on the Quality of English LanguageThere is need for English language editing to make the work better
Author Response
Dear Reviewer:
Thanks to the reviewers for their comments and suggestions. I will comment or make changes based on your suggestions. You can see the attached “ijms-2818772-paper” with the revisions highlighted in blue.
1.The overall structure of the manuscript is confusing. The methodology should come before the results to promote coherence.
Response 1:Thanks a lot for your comments. We adjusted the overall structure of the manuscript and put methods before results to make the structure more organized.
2.The methodology is not very elaborately explained and the authors are not clear on the various sources of the methodology used. The authors should acknowledge the sources or if they have invented them also say so.
Response 2:Thanks a lot for your comments. Many of our methods are tested according to the reagent manufacturer's plan, and the specific experimental procedures have been supplemented.
LINE113-115:
After washing treated sequencing cell samples 3 times with PBS, total RNA was isolated from co-infected A549 cells using the TRIzol™ Plus RNA Purification Kit (Thermo Fisher Scientific, Carlsbad, CA USA) according to the manufacturer's instruc-tions.
LINE120-125:
The library is amplified and constructed through the steps of purifying and frag-menting mRNA, synthesizing and purifying double-stranded cDNA, end repair/dA tail addition, ligation of adapters, purification of ligation products and amplification. Use gel electrophoresis to detect the library. The detected library is between 300-500bp and is considered a qualified library. Finally, the qualified libraries were subjected to paired-end sequencing using Illumina HiSeq4000 (BGI, Shenzhen, China).
LINE127-130:
Before alignment, FastQC was used to check the quality of the raw reads gener-ated by the Illumina Hiseq 4000 platform that were filtered by removing the dirty raw reads. Reads containing adapter, an unknown base >10%, and low-quality reads (The base number of threshold mass ≤10 accounts for more than 50% of the total reading) were removed to obtain clean reads of mRNA.
3.The results are not well sorted and appear very crowded, the authors should present the key results well elaborated and the rest should be appendices.
Response 3:Thanks a lot for your comments. We optimised some of the results and put them into the Supplementary Material.
4.The discussion though well crafted is sometimes confusing, for example line 385-386 ''This suggests that as adenoviral infection worsens, S. aureus can also be stressed can also cause the release of complement factors such as CFH and CFB in the epithelial cells, which inhibit the over-activation of C3 to allow S. aureus to evade the immune response'' it is not clear whether the implication is that it this process inhibits or promotes S. aureus' evasion of the immune system. Please as you are doing English translation or editing, take care to retain the meaning.
Response 4:Thanks a lot for your comments. We have optimized the controversial parts of the conclusion to make it scientifically sound.
LINE 418-422:
This indicates that as adenoviral infection progresses, S. aureus may experience stress and release complement factors like CFH and CFB in epithelial cells to prevent over-activation of C3 and evade the immune response. [32,33] Moreover, S. aureus auto-protein Eap can block adhesion factors such as ICAM-1, inhibiting leukocyte recruit-ment [33].

Reviewer 3 Report
Comments and Suggestions for Authors
The paper titled, "Activation of the RIG-I/MAVS signaling pathway during human adenovirus type 3 infection impairs the proinflammatory response induced by secondary infection with Staphylococcus aureus".
The authors used an A549 cell model and demonstrated that HAdV-3E promotes S. aureus infection by affecting epithelial cells of the lungs. The study reveals an unknown mechanism of action in the secondary infection by HAdV-3E and S. aureus, contributing to a better understanding of the pathogenesis involved. Overall, the study has potential and the methods and results are articulated. However, I have some minor comments for the authors to address.
Minor comments
Line 27: Add "a non-enveloped" instead of "envelope-less"
Line 56, I think the authors should mention the full name of S. aureus, the first time to introduce the name in the manuscript. Same line, "Gram" in place of "gram". Also, put a space between the period (.) and aureus. Kindly check for this throughout the manuscript.
Line 66, mention the full name of "MRSA".
Line 69-71, kindly provide a reference for the information.
LIne 85-87 should have a reference. Also, there is a space between the lines of the paragraph, kindly check.
Line 99, I don't know what the authors mean "We will find that some of the 24h coinfection group (A3SA_24), have developed significant lesions, rounding and necrosis in the bright field, but no aggregation has occurred"
All the results should be explained in the past tense. Kindly modify the results accordingly.
Line 119, it should be "analysis"
In the sentence "A3SA_48 compared to A3SA_24 compared to A3SA_48 compared to A3SA_36 and A3SA_36 compared to A3SA_24 there was," there seems to be repetition and confusion. For more clarity, I would recommend the authors to rephrase it.
Kindly cross-check for FigA, it seems like a typo.
Line 350, italicize Mycobacterium tuberculosis.
Comments on the Quality of English Language
Minor modifications are needed.
Author Response
Dear Reviewer:
Thanks to the reviewers for their comments and suggestions. I will comment or make changes based on your suggestions. You can see the attached “ijms-2818772-papaer” with the revisions highlighted in blue.
Line 27: Add "a non-enveloped" instead of "envelope-less"
Response 1:Thanks a lot for your comments. I have updated it.
LINE 27-28:
Human adenovirus is a non-enveloped, double-stranded DNA virus that was first isolated in 1953 from surgically removed glands of children [1].
Line 56, I think the authors should mention the full name of S. aureus, the first time to introduce the name in the manuscript. Same line, "Gram" in place of "gram". Also, put a space between the period (.) and aureus. Kindly check for this throughout the manuscript.
Response 2:Thanks a lot for your comments. I have updated it.
LINE 56-58:
Staphylococcus aureus (S.aureus) is a common Gram-positive pathogen commonly secondary to viral pneumonia infections. It was first observed in 1880 in the pus of a surgical abscess [15]. Soon after the discovery of S.aureus , it was quickly realized that S.aureus is a powerful pathogen and remains one of the leading causes of bacterial in-fections globally [16].
Line 66, mention the full name of "MRSA".
Response 3:Thanks a lot for your comments. I have updated it.
LINE 64-47:
A retrospective cohort study showed 17% of intensive care unit patients had positive nasal swabs for Methicillin-resistant Staphylococcus aureus (MRSA) and 28.6% of patients with nasal MRSA nasal colonization subsequently developed pneumonia [19].
Line 69-71, kindly provide a reference for the information.
Response 4: Thanks a lot for your comments. I added reference 21
21.Zhu, Z.; Hu, Z.; Li, S.; Fang, R.; Ono, H. K.; Hu, D.-L. Molecular Characteristics and Pathogenicity of Staphy-lococcus Aureus Exotoxins. IJMS 2023, 25 (1), 395. https://doi.org/10.3390/ijms25010395.
LIne 85-87 should have a reference. Also, there is a space between the lines of the paragraph, kindly check.
Response 5:Thanks a lot for your comments. Yes, I have corrected the content and supplemented Ref. 24
LINE85-87:
A surveillance of age-specific community-acquired pneumonia coinfections in China from 2009-2020 showed that the occurrence of community-acquired pneumonia in children and adolescents was significantly associated with adenoviral and S.aureus coinfections [24].
24.Liu, Y.-N.; Zhang, Y.-F.; Xu, Q.; Qiu, Y.; Lu, Q.-B.; Wang, T.; Zhang, X.-A.; Lin, S.-H.; Lv, C.-L.; Jiang, B.-G.; et al. Infection and Co-Infection Patterns of Community-Acquired Pneumonia in Patients of Different Ages in China from 2009 to 2020: A National Surveillance Study. The Lancet Microbe 2023, 4, e330–e339, doi:10.1016/S2666-5247(23)00031-9.
Line 99, I don't know what the authors mean "We will find that some of the 24h coinfection group (A3SA_24), have developed significant lesions, rounding and necrosis in the bright field, but no aggregation has occurred"
Response 6:Thanks a lot for your comments. We've made some modifications.
LINE168-172:
Results showed that after 24h of coinfection, some cells displayed significant lesions and rounding in bright field, although no aggregation was observed. Specifically, the 24h coinfection group (A3SA_24) exhibited partially rounded lesions under bright field, with 20% of epithelial cells expressing green fluorescence indicative of infection.
All the results should be explained in the past tense. Kindly modify the results accordingly.
Response 7:Thanks a lot for your comments. We've made some modifications.
Line 119, it should be "analysis"
In the sentence "A3SA_48 compared to A3SA_24 compared to A3SA_48 compared to A3SA_36 and A3SA_36 compared to A3SA_24 there was," there seems to be repetition and confusion. For more clarity, I would recommend the authors to rephrase it.
Kindly cross-check for FigA, it seems like a typo.
Response 8:Thanks a lot for your comments. I have updated it.
LINE 198-200:
The results of volcano plots were analyzed to compare the differential gene ex-pression between A3SA_48 and A3SA_24, A3SA_48 and A3SA_36, as well as A3SA_36 and A3SA_24.
Line 350, italicize Mycobacterium tuberculosis.
Response 9:Thanks a lot for your comments. I have updated it.
LINE 384-386:
For example, upregulation of RSAD2 in Mycobacterium tuberculosis hinders host defense activation and antigen presentation in dendritic cells during infection with this pathogen [28].

Round 2
Reviewer 1 Report
Comments and Suggestions for Authors
The authors have not significantly improved the quality of their writing; confusing descriptions persist, and the results remain poorly presented, impeding comprehension.
The purpose of the manuscript remains unclear, marked by a noticeable absence of explicit hypotheses. It appears that the authors intend to investigate whether prolonged HAdV-3E infection increases susceptibility to secondary S. aureus infections. However, crucial experiments assessing the effects of the duration of HAdV-3E infection on S. aureus internalization, intracellular survival, and persistence were omitted. The absence of these findings renders the overall work nonsensical. Indeed, it seems improbable that the duration of HAdV-3E infection plays a determining role in secondary S. aureus infections. Given this, it is challenging for this reviewer to understand the rationale behind the experimental design.
Overall, the experimental design lacks scientific rigor without a substantial purpose, making it unlikely to derive a robust conclusion about the restriction of secondary S. aureus infections by the RIG-I signal pathway target. It is recommended that the authors conduct additional experiments in their future studies to ascertain whether targeting the RIG-I signal pathway is sufficient to limit S. aureus infection following HAdV-3E infection.
L16-17: The sentence is incomplete...grammar error.
L18-21: The model used involved the infection of cells with HAdV-3E, followed by exposure to S. aureus. However, it is important to clarify that the primary cause is HAdV-3E, with S. aureus acting as a secondary factor subsequent to HAdV-3E infection. Please review the manuscript to rectify this discrepancy.
L22-26: The expression is very confusing…rewrite.
L26-27: The conclusion is not succinct and lacks clarity. Please ensure that the concluding statements are direct and to the point.
L39-42: Re-write
L43: Should it be “during epidemics or outbreaks”?
L94-97: The description is confusing. Why did you focus on RIG-I?
………
L103-105: Confusing…
L112: Why did you use 2% FBS for cell culture?
L113-114: How did you know that each well had 5×105 cells? Was the initial cell count not conducted?
L141: |FoldChange|>0.2?
L148: What is the significance of "34s" in this context?
Comments on the Quality of English LanguageQuality of English language must be improved.
Author Response
The authors have not significantly improved the quality of their writing; confusing descriptions persist, and the results remain poorly presented, impeding comprehension.
Response1:Thank you for your comment. We are already trying our best to improve our English and make articles easier to understand.
The purpose of the manuscript remains unclear, marked by a noticeable absence of explicit hypotheses. It appears that the authors intend to investigate whether prolonged HAdV-3E infection increases susceptibility to secondary S. aureus infections. However, crucial experiments assessing the effects of the duration of HAdV-3E infection on S. aureus internalization, intracellular survival, and persistence were omitted. The absence of these findings renders the overall work nonsensical. Indeed, it seems improbable that the duration of HAdV-3E infection plays a determining role in secondary S. aureus infections. Given this, it is challenging for this reviewer to understand the rationale behind the experimental design.
Response2:Thank you for your comment. It is possible that the statement in the article may need further clarification. Our primary focus is to demonstrate the significant role of secondary Staphylococcus aureus in the overall infection process. Clinical evidence supports the notion that secondary bacterial infections are crucial in the progression of infections. Consequently, our research primarily investigates how prolonged HAdV-3E infection can impair the activation of certain inflammatory pathways within cells, consequently facilitating secondary Staphylococcus aureus infections.
To do this study, we really need to explore the duration of HAdV-3E infection and the concentration of Staphylococcus aureus on cell viability. We designed continuous infection with HAdV-3E with different multiplicity of infection and then secondary infection with Staphylococcus aureus with different multiplicity of infection for 6 hrs. , 9hrs, and 12hrs infection models to evaluate the impact of HAdV-3E infection duration on Staphylococcus aureus internalization, intracellular survival, and persistence.
LINE171-178:
A continuous infection model with HAdV-3E at varying MOI followed by sec-ondary infection with S.aureus at different MOI for durations of 6hrs, 9hrs, and 12hrs revealed that cell survival rates were notably low at 9hrs and 12hrs post high-concentration S.aureus infection, both remaining below 60% with significant cell damage. Persistent 6hr infection by S.aureus resulted in reduced cell survival compared to single virus infection, yet achieved a relatively high survival rate of 60%, ensuring the persistence of S.aureus infection. Consequently, a 6-hour secondary infection model with S.aureus was selected for experimental purposes.( Supplementary Fig 1 Sup-plementary Fig 2 Supplementary Fig 3)
Overall, the experimental design lacks scientific rigor without a substantial purpose, making it unlikely to derive a robust conclusion about the restriction of secondary S. aureus infections by the RIG-I signal pathway target. It is recommended that the authors conduct additional experiments in their future studies to ascertain whether targeting the RIG-I signal pathway is sufficient to limit S. aureus infection following HAdV-3E infection.
Response3:Thank you for your feedback. Our team is currently focusing on further investigating RIG-I target restriction to demonstrate how targeting the RIG-I signaling pathway could potentially enhance Staphylococcus aureus infection. The upcoming experiments will be detailed in our next manuscript.
L16-17: The sentence is incomplete...grammar error.
Response4:Thank you for your comment. We've improved it
LINE15-18:
The influence of host-pathogen interactions on the progression of the disease remains unclear. It is noteworthy that HAdV-3E secondary to S. aureus infection is frequently observed in clinical settings epidemiologically, yet the underlying mechanism of susceptibility remains unidentified.
L18-21: The model used involved the infection of cells with HAdV-3E, followed by exposure to S. aureus. However, it is important to clarify that the primary cause is HAdV-3E, with S. aureus acting as a secondary factor subsequent to HAdV-3E infection. Please review the manuscript to rectify this discrepancy.
Response5:Thank you for your comment. We've improved it
LINE18-22
This study utilized an A549 cell model to investigate secondary infection with S.aureus following HAdV-3E infection. The findings suggest that HAdV-3E exacerbates S.aureus infection by inten-sifying lung epithelial cell damage. The results highlight the role of HAdV-3E in enhancing the interferon signaling pathway through RIG-I (DDX58), resulting in increased expression of inter-feron-stimulating factors like MX1, RSAD2, and USP18.
L22-26: The expression is very confusing…rewrite.
Response6:Thank you for your comment. We've improved it
LINE23-24
The increase in interferon-stimulating factors inhibits the NF-κB and MAPK/P38 pro-inflammatory signaling pathways.
L26-27: The conclusion is not succinct and lacks clarity. Please ensure that the concluding statements are direct and to the point.
Response7:Thank you for your comment. We've improved it
LINE24-26
These findings reveal new mechanisms of action of HAdV-3E and S.aureus in secondary infections, enhancing our comprehension of pathogenesis.
L39-42: Re-write
Response8:Thank you for your comment. We've improved it
LINE37-42
These viruses can readily cause lung infections in the elderly and children, particu-larly in children under 5 years of age who are more vulnerable to human adenovirus infection, leading to severe pneumonia [4,5]. Adenovirus pneumonia has been docu-mented to represent around 10% of childhood pneumonias, and it can spread easily, sometimes resulting in fatal illness in healthy children during epidemics or outbreaks [6]
L43: Should it be “during epidemics or outbreaks”?
Response9:Thank you for your comment. We've improved it
LINE43-45
During epidemics or outbreaks, Adenoviruses can lead to necrotizing bronchiolitis and fine bronchiolitis with extensive exfoliation of the surface epithelium. This is typically observed following viral infection of the lungs, particularly in medium-sized intrap-ulmonary bronchi [7-9].
L94-97: The description is confusing. Why did you focus on RIG-I?
Response10:Thank you for your comment. We've improved it
LINE93-98
We examined the interplay between adenovirus and S. aureus co-infection using high-throughput data in a cell model of secondary infection with S. aureus after HAdV-3E infection for varying durations. Our findings revealed the significance of RIG-I targets in bacterial infections. This study lays the groundwork for potential drug development targeting RIG-I for adenoviral-bacterial coinfections.
L103-105: Confusing…
Response11:Thank you for your comment. The adenovirus should be changed to HAdV-3E. This article primarily focuses on the amplification and culture of HAdV-3E in A549 cells. The A549 cells were sourced from Manassas, VA, USA and were maintained in the laboratory.
L112: Why did you use 2% FBS for cell culture?
Response12:Thank you for your comment. To ensure that A549 cells can survive and not detach from the bottom of the cell flask for more than 48 hours, a certain amount of FBS is required, so 2% FBS was added.
L113-114: How did you know that each well had 5×105 cells? Was the initial cell count not conducted?
Response13:Thank you for your comment. Yes, I started by using a hemocytometer for cell counting.
L141: |FoldChange|>0.2?
Response14:Thank you for your comment. This is a clerical error, it should be |FoldChange|>2, it has been corrected
L148: What is the significance of "34s" in this context?
Response15:Thank you for your comment. This is the annealing and extension step of fluorescence quantitative PCR.

Round 3
Reviewer 1 Report
Comments and Suggestions for Authors
I recommend that the authors provide quantitative data on the internalization and proliferation of S. aureus in HAdV-3E-infected cells under the specified conditions, along with quantitative data on specified mode of cell death rather than only descriptive accounts. Additionally, it is evident that there is a lack of essential experimental groups where cells are infected with HAdV-3E or S. aureus alone, which should be included for comparison. Furthermore, the authors should validate the role of the RIG-I pathway in the context of S. aureus infection secondary to HAdV-3E.
The Introduction part must be reorganized to clearly state the hypothesis and objectives that align with a coherent background. Moreover, the data presentation and interpretation must be enhanced.
Comments on the Quality of English LanguageExtensive editing of English language required.
Author Response
I recommend that the authors provide quantitative data on the internalization and proliferation of S. aureus in HAdV-3E-infected cells under the specified conditions, along with quantitative data on specified mode of cell death rather than only descriptive accounts.
Response1: Thank you very much for your comments. In this study, the quantitative data of the cell survival rates of Staphylococcus aureus models of 6h, 9h and 12h secondary infection with different MOI after infection with HAdV-3E of different MOI for 48 hours are presented in the form of histograms (Fig .2 Supplementary Fig 1 Supplementary Fig 2).
Additionally, it is evident that there is a lack of essential experimental groups where cells are infected with HAdV-3E or S. aureus alone, which should be included for comparison.
Response2:The bar graph in FIG.2 displays the basic experimental group infected with HAdV-3E or S. aureus alone. In Fig.2, * (P < 0.05) and *** (P < 0.001) indicate that cell activity is higher when infected with HAdV-3E or Staphylococcus aureus alone compared to the co-infected group. This suggests that the damage in the co-infected group was greater than in the single-infection groups, aligning with real clinical scenarios and forming the basis for the cell model.
Furthermore, the authors should validate the role of the RIG-I pathway in the context of S. aureus infection secondary to HAdV-3E.
Response3:Thank you very much for your suggestion. This issue is very important and we will conduct further research on this issue in the future.
The Introduction part must be reorganized to clearly state the hypothesis and objectives that align with a coherent background. Moreover, the data presentation and interpretation must be enhanced.
Response4:Thank you very much for your suggestion. We have reorganized the introduction, especially the third paragraph, to reflect the reasons and purposes of this study. At the same time, we also explain some of the data in the introduction. In LINE 49-51, we supplemented S. aureus to elicit the co-infection phenomenon of adenovirus and S. aureus. Add sentences summarizing the previous text in LINE71-73. The clinical phenomenon of adenovirus-Staphylococcus aureus co-infection introduced in LINE89-101 illustrates the importance of studying the mechanism of adenovirus-Staphylococcus aureus co-infection. and explained the main purpose and significance of this study.
LINE 49-51:
Additionally, viruses can lead to dysregulation of various components of the immune system, both pro- and anti-inflammatory, thereby promoting pathogenesis by oppor-tunistic pathogens such as S. aureus.
LINE 71-73:
Hence, the likelihood of secondary infection with S.aureus is notably high among children and individuals with compromised immune systems.
LINE 89-101:
Children with adenovirus infection in their respiratory tract often experience co-infection with S.aureus, leading to severe pneumonia. A surveillance report on community-acquired pneumonia co-infections across different age groups in China from 2009 to 2020 highlighted the significant association between community-acquired pneumonia in children and adolescents and co-infection with adenovirus and S.aureus [24]. While there has been progress in understanding the molecular mechanisms of ad-enovirus and S.aureus infections individually, the molecular mechanism of pneumonia resulting from secondary S.aureus infection following adenovirus infection remains poorly understood. To address this gap, we established a co-infection cell model of HAdV-3E and S.aureus and utilized high-throughput sequencing methods to elucidate the interaction between adenovirus and S.aureus on lung epithelial cells. Our study identified the importance of RIG-I targets in bacterial infections, laying the foundation for potential drug development targeting RIG-I in adenoviral-bacterial coinfections.
